# The Pig as a Translational Animal Model for Biobehavioral and Neurotrauma Research

**DOI:** 10.3390/biomedicines11082165

**Published:** 2023-08-01

**Authors:** Alesa H. Netzley, Galit Pelled

**Affiliations:** 1Department of Biomedical Engineering, Michigan State University, East Lansing, MI 48824, USA; hughson3@msu.edu; 2Neuroscience Program, Michigan State University, East Lansing, MI 48824, USA; 3Department of Mechanical Engineering, Michigan State University, East Lansing, MI 48824, USA; 4Department of Radiology, Michigan State University, East Lansing, MI 48824, USA

**Keywords:** neurotrauma, traumatic brain injury, translational neuroscience, large animal, behavior, swine, pig, cognition, physiology

## Abstract

In recent decades, the pig has attracted considerable attention as an important intermediary model animal in translational biobehavioral research due to major similarities between pig and human neuroanatomy, physiology, and behavior. As a result, there is growing interest in using pigs to model many human neurological conditions and injuries. Pigs are highly intelligent and are capable of performing a wide range of behaviors, which can provide valuable insight into the effects of various neurological disease states. One area in which the pig has emerged as a particularly relevant model species is in the realm of neurotrauma research. Indeed, the number of investigators developing injury models and assessing treatment options in pigs is ever-expanding. In this review, we examine the use of pigs for cognitive and behavioral research as well as some commonly used physiological assessment methods. We also discuss the current usage of pigs as a model for the study of traumatic brain injury. We conclude that the pig is a valuable animal species for studying cognition and the physiological effect of disease, and it has the potential to contribute to the development of new treatments and therapies for human neurological and psychiatric disorders.

## 1. Introduction

Translational neuroscience within biomedical research is a rapidly growing field, aiming to bridge the gap between basic science and clinical applications by using the findings from preclinical studies to develop more effective interventions and treatments for human neurological disorders and injuries [1,2]. The use of animal models is an essential step in the research process as these models provide a means to study the underlying mechanisms behind various disease states and the holistic effects of interventions in a living system [3]. The development of animal models that accurately replicate aspects of human neurological conditions is key to the success of translational neuroscience research. In this regard, pigs (sus scrofa) have emerged as a highly valuable model species due to notable similarities between pig and human neuroanatomy, physiology, and behavior [4,5,6,7].

Over the past several decades, the pig has gained significant attention as a model animal for translational and biobehavioral research, particularly in the fields of neurotrauma and cognitive neuroscience. The relative cognitive complexity, social behavior, and overall body composition of the pig make it an exceptional model organism for studying various disease states [8,9]. Pigs have been used extensively in studies of pharmacology and toxicology as a major translational model due to their vast biochemical similarities to humans [10]. Another important aspect of biomedical engineering is the development of new equipment and testing the diagnostic and prognostic potential of new methods and therapies. The pig is instrumental in this regard, as its large body size allows the use of equipment and modalities with results that can be immediately applicable to humans [11]. 

This review will examine the different ways in which pigs are used to study cognition and behavior. We will highlight many commonly used behavioral tasks that pigs are capable of performing as well as several contemporary physiological assessments. We will also highlight the use of pigs for the study of traumatic brain injury as many pig models of brain injury have been developed, as well as how these models contribute to our understanding of brain function and dysfunction in humans. Finally, we will discuss the future directions of pig-based translational biobehavioral research, including the development of advanced techniques for assessing pig behavior, and the potential of pig models for studying novel treatments for neural injuries and neuropsychiatric disorders. 

## 2. Review

### 2.1. Preclinical Research

Preclinical research is an essential step in advancing our understanding of neurological disorders and developing effective treatments. Animal models play a crucial role in preclinical neuroscience, providing valuable insights into the underlying mechanisms of brain function and dysfunction in a living system. There are several advantages to using animal models, including the ability to control experimental variables, genetic manipulations, perform invasive procedures, and analyze biological samples in ways that cannot be done in human subjects [12]. Over the past century, various animal species have been used to model diseases and conditions in neuroscience research [13]. Of these various animal model species, the most common and well-characterized are rodents, specifically mice and rats [14].

Rodent models have greatly advanced our understanding of fundamental neurological processes. Notably, rodent research has contributed to the study of neuronal function, synaptic plasticity, neurotransmission, and connectivity [15,16]. Rats and mice have been used extensively to model a wide range of neurological and psychiatric disorders, including neural injury, degenerative diseases, and psychiatric conditions such as depression, anxiety, and drug addiction [17]. By replicating these disorder-related conditions and phenotypes, researchers can investigate the underlying mechanisms behind the disorders, study disease progression, and develop and test the efficacy of potential therapeutic interventions. Preclinical rodent research continues to play a critical role in the development and evaluation of novel therapeutic interventions. Rodent models have been used to test pharmaceutical interventions, gene therapies, and various other treatments, with many showing great promise [18]. Rodents and humans share many physiological and genetic similarities, thus rodent studies allow for rigorous preclinical testing of interventional strategies for neurological disorders before moving on to human trials. 

### 2.2. Translational Considerations

The use of rodent models in preclinical research has contributed significantly to the development of a wide variety of treatments for various neurological disorders. Although many potential treatments have shown promise in rodents, the vast majority of the therapeutics tested have failed in clinical trials [19,20]. Many researchers speculate that one major contributing factor toward the low rate of translational success is due to the anatomical and physiological differences between humans and rodents [6]. The anatomical differences between rodent and human brains are considerable. Rats and mice have small lissencephalic brains with a low white to gray matter ratio [6]. In this regard, rodents may be a less-than-ideal model organism for the study of diseases that have a major effect on white matter regions, such as Multiple Sclerosis [21]. Additionally, the lack of cerebral convolutions can be problematic for studies of brain injury as stress generated by mechanical insult tends to be distributed more evenly across surface areas in smooth brains [22]. In contrast, the presence of gyri and sulci in the human brain focuses mechanical stress toward the base of the sulci, deeper into the center of the brain and in close proximity to white matter regions [22]. 

While the overall organization of brain structures is fairly similar between primates and rodents, investigators are learning that the functions of these brain regions may not be as alike as previously thought [23]. A recent report by Hodge et al. (2019) has demonstrated considerable differences in the expression of genes among similar cell types within mouse and human brains [24]. In this study, researchers used single-cell transcriptomics to characterize various cell types in the cortex of mice and the human middle temporal gyrus. They then compared the expression of genes in homologous cell types and found that among similar cell types, there were major divergencies in the expression of neurotransmitter receptors, ion channels, extracellular matrix elements, and cell-adhesion molecules between species [24]. This divergent expression of key signaling elements could help to provide some justification as to why the translation rate of pharmacological treatments for neurological disorders has been so low. 

### 2.3. Similarities of the Pig Brain

The pig brain shares many structural and functional similarities with the human brain, including comparable cortical organization, anatomical structure (as shown in Figure 1), gray and white matter ratio, and regional distribution of neurotransmitter systems [25,26]. They have a similar brain size and structure to humans, with analogous cortical folding, white matter tracts, and subcortical structures [27,28]. Additionally, pigs exhibit similar patterns of brain development, allowing researchers to investigate the underlying neurodevelopmental processes associated with developmental disorders [29]. 

The pig brain undergoes a period of rapid growth and development during the perinatal period, which is comparable to human brain development during late gestation and early infancy [25,30,31]. Another important similarity to humans is the chronological development pattern of the neocortex; comparative studies demonstrated that the neurogenesis of a domestic pig is completed before term [29]. Additionally, imaging studies using Diffusion Tensor Imaging (DTI) have demonstrated that the myelination rate in the corpus callosum of pigs is similar to humans [25]. Pigs also exhibit many human-relevant behaviors, such as social structure, enabling the investigation of complex cognitive functions that are often impaired in human neurological disorders such as learning, memory, and social behavior [32,33,34]. Moreover, the pig’s physiology and metabolism are closer to humans compared to smaller animal models, enabling a more accurate evaluation of pharmacokinetics and treatment responses [35,36]. Collectively, these factors make the pig a valuable model for studying human neurological conditions and injury, facilitating the translation of preclinical findings to clinical applications, and enhancing our understanding of these complex disorders.

### 2.4. Pig Cognition and Behavior

Behavioral assessment in preclinical research provides a bridge between basic neuroscience and clinical applications. The study of cognition and behavior is essential for understanding brain function and modeling human disorders. Many neurological and psychiatric disorders manifest as cognitive and behavioral impairments. By observing and quantifying behavioral responses in animal models, researchers can investigate various aspects of brain function, including sensory perception, motor control, learning and memory, social behavior, and emotional processes. Pigs possess a high level of cognitive complexity, exhibiting a wide range of behaviors and mental abilities that are relevant to studying human cognition. Pigs demonstrate advanced social behavior, problem-solving skills, spatial memory, and learning capabilities. In order to use pigs as a model animal for biobehavioral research, it is important to have reliable methods for assessing their behavior. Direct observation and scoring are commonly used, as well as video recording and computerized analysis, which enable researchers to monitor and measure behavioral parameters with high precision and accuracy. Automated systems, such as accelerometers and radio-frequency identification (RFID), are also being developed for monitoring pig behavior in a non-invasive manner. The advancement of technology continuously facilitates the development of sophisticated and precise behavioral assessment methods. 

#### 2.4.1. Spatial Memory and Maze Tests

Spatial memory deficits are observed in various neurological disorders. Maze tests can be used to assess spatial learning and memory in pigs. Pigs are trained to navigate through mazes, such as T-mazes or radial arm mazes, to locate rewards or escape routes. The T-maze apparatus consists of a central stem and two perpendicular arms, forming a T-shape. The stem is typically a long corridor or runway, while the arms are shorter and lead to distinct goal areas that customarily have a food reward. Similarly, a radial arm maze consists of a central hub and several radiating arms, typically eight, arranged in a circular pattern. Performance in these tests can reveal a pig’s ability to remember and utilize spatial information. A T-maze was used to evaluate a pig’s spatial cognition after brain injury, published by Kinder et al. (2019). They found that pigs with a brain injury required more time to make a decision and were less accurate when deciding which arm of the maze would contain a reward [37]. Similarly, Singh et al. (2019) found that pigs with hypoxia-ischemia (HI) performed more slowly in a T-maze compared to control pigs, suggesting that HI pigs have poorer working memory [38]. While less common, the eight-arm radial maze has also been used to assess spatial cognition in pigs. In a study published by Chen et al. (2021), neonatal pigs whose diets were supplemented with lactoferrin were able to reach learning objectives in the radial maze faster than the experimental controls [39]. These studies show that maze tasks are valuable assessment tools for their use with pigs. 

The spatial hole board test is another commonly used assessment to test spatial learning and working memory in pigs [40,41]. The apparatus typically consists of a rectangular platform with evenly spaced holes or containers. During testing, food rewards or pellets are placed into some of the holes, while other holes are left empty. Pigs are tasked with retrieving the food in the most efficient way possible without returning to holes which have already been explored. Performance can be measured by the number of correct choices, the latency to find rewards, and the pattern of exploration. A recent study by Clouard et al. (2022) reported that pigs who were fed oligosaccharides spent more time in between visits to different baited buckets, which resulted in higher scores for memory and fewer errors. The researchers concluded that the pigs who were fed oligosaccharides demonstrated higher executive functioning than the controls who exhibited more sporadic, hyperactive behavior [42]. 

A spatial baited ball pit task was first published by Netzley et al. (2021), wherein food rewards were distributed evenly throughout a shallow pool and hidden under colorful plastic balls (Figure 2). The investigators found that over the course of several weeks, healthy pigs become faster and more accurate at retrieving the rewards [43]. 

#### 2.4.2. Object Recognition and Discrimination Tests

Object recognition tests can be used to assess the ability of pigs to discriminate between colors, shapes, or familiar and novel objects. These tasks test recognition memory, and often, the training for these tasks involves familiarizing the animal with a certain object and later presenting the trained object alongside one or more new or incorrect objects. For example, a three-choice color discrimination task was used to assess recognition memory in minipigs by Schramke et al. (2016). Pigs were presented with three colored boxes (blue, red, and yellow), each containing a food reward. Pigs were trained to recognize that only the blue box could be opened, while the red and yellow boxes were sealed. This task was given to both healthy pigs and pigs modeling Huntington’s Disease, although no differences in task performance between groups were found [44]. Ao et al. (2022) developed a rack-mounted touchscreen device for pigs given a color discrimination task. Briefly, a touchscreen monitor was placed on an audio rack, and an automatic pellet dispenser was positioned across the room. Pigs were tasked with snout-touching a colored shape on the screen, and successful touches were reinforced with an audio cue while a fruit-flavored pellet was deposited from the dispenser [45].

#### 2.4.3. Social Interaction and Aggression Tasks

Social interaction tests are used to evaluate a pig’s social behavior and cognition. They often involve introducing a subject to an unfamiliar pig in order to observe social interactions between animals, dominance hierarchies, play behavior, and social recognition. A voluntary human approach test was conducted by Wegner et al. (2020), which revealed that pigs who approached a motionless human observer more quickly were more likely to engage in forceful means of contact, i.e., biting, when compared to pigs who were more reserved [46]. Aggression tests assess aggressive behavior and response to social challenges in pigs. In these tests, pigs are introduced to stimuli or provocations that elicit aggressive responses, such as competing for resources or territorial intrusions. One example of a dominance task was published by Schramke et al. (2016). Two pigs were led into a tunnel separated by a trap door. The trap door was removed, and the more dominant pig pushed past the less dominant one in order to reach a food reward given behind the opposing pig [44]. 

#### 2.4.4. Vocalization Analysis 

Vocalization analysis involves recording and analyzing the vocalizations emitted by pigs in different contexts. It provides insights into their communicative behaviors, emotional states, and responses to stimuli. Pigs are a vocal species, who communicate in a herd by calling to one another. A speech recognition paradigm for pigs was recently developed by Wu et al. (2022), utilizing a fusion network that combines both spectral and audio features to classify individual pig speech patterns [47]. Vocal expression has also been used to assess emotional states in pigs. In a study by Briefer et al. (2022), pig vocalizations were used to develop an automated vocal recognition system to monitor animal welfare in a farm setting [48].

#### 2.4.5. Fear and Anxiety Tasks

Fear and anxiety tests assess an animal’s responses to fear-inducing stimuli, such as open fields, sudden loud sounds, or even unfamiliar objects and locations. Observing behaviors like freezing, avoidance, or elevated stress markers can provide insights into porcine emotional states and stress responses. The open field test is a widely used behavioral test to assess exploratory behavior, locomotor activity, and anxiety-like responses in animals including pigs. The test is traditionally conducted in a square or rectangular arena, usually made of an open and brightly lit area, devoid of any specific cues or obstacles. The animal is placed in the center of the arena, and their behavior is recorded and analyzed. In pigs, typical behavioral parameters include the distance traveled, exploration of different zones such as the center or periphery of the arena, and specific behaviors such as rearing, sniffing, and defecation [49]. These measures provide insights into the animal’s exploratory behavior, as well as anxiety levels. For example, greater distance traveled, and more time spent in the center of the arena are indicative of reduced levels of anxiety. On the other hand, freezing, defecating, and staying near the walls of the arena may indicate higher anxiety levels (Figure 3) [43,50]. In a study published by Haigh et al. (2020), it was determined that pigs who were bitten by other members of the herd exhibited more anxious behaviors in the open field compared to pigs who committed the biting acts [49]. Overall, the open field test is a well-established and widely used method to evaluate behavior in animals, including pigs, and has contributed significantly to our understanding of anxiety-related behaviors and exploratory tendencies. 

Behavioral assessment is a fundamental component of preclinical neuroscience research, providing valuable information about the functioning of the nervous system and its relationship to behavior. It plays a crucial role in the development and evaluation of potential treatments for neurological and psychiatric disorders, ultimately contributing to the advancement of clinical neuroscience. Table 1 (below) lists several key behavioral assessments conducted in pigs and the various experimental conditions that have been evaluated. 

### 2.5. Physiological Assessments

Preclinical researchers utilize physiological assessments to gain a deeper understanding of the underlying biological mechanisms that drive changes in cognition and behavior. These assessment methods provide valuable data on various physiological parameters, such as heart rate, blood pressure, respiration rate, hormonal levels, and neural activity. Physiological assessments can be particularly important in preclinical research because they provide objective and quantitative measurements of biological processes that may not be directly observable through behavioral assessments alone. These assessments also help researchers investigate the effects of experimental manipulations on the function of different organ systems, including the cardiovascular, respiratory, endocrine, and nervous systems. For example, researchers can use electroencephalography (EEG) or functional magnetic resonance imaging (fMRI) to study brain activity patterns and neural responses. They can measure neurotransmitter levels or hormone concentrations to examine the impact of experimental interventions on biochemical signaling. Physiological assessments also enable researchers to monitor vital signs and physiological parameters during different experimental conditions, helping to ensure animal welfare and safety during the study.

#### 2.5.1. Real-Time Physiological Monitoring

As technology advances, more accessible physiological monitoring devices have been developed. Pigs have been equipped with many of the technologies used to monitor health and wellness in humans. Wearable electrocardiograms have been used to monitor heart rate and blood pressure in pigs in a study published by Nachman et al. (2020). They found that the wearable device was able to consistently monitor these parameters in pigs with hemorrhagic shock despite unstable physiological conditions [51]. Other wearable devices have been used to monitor the health of pigs in research. Healthy minipigs were outfitted with human Fitbit^®^ devices in Netzley et al. (2021) in order to track their activity during the course of a 12 h day, finding that the pigs were most active between the hours of 12 pm and 4 pm [43]. While wearable devices are desirable for human patients, pigs are curious and often chew anything they can reach, thus some studies may facilitate a need for implantable monitoring equipment. Martinez-Ramirez et al. (2022) demonstrated the capability of subdural implanted EEG monitoring systems for the long-term assessment of post-traumatic epilepsy in freely ambulating pigs for up to 13 months [52]. As a whole, pigs provide a fantastic model in which to test the validity of various novel devices or to use pre-established technologies for comparison to human data. 

#### 2.5.2. Neuroimaging

Neuroimaging plays a crucial role in translational research by providing valuable insights into the structure, function, and connectivity of the brain. Imaging studies can bridge the gap between preclinical studies and clinical research by facilitating the translation of findings from the laboratory to real-world applications. Neuroimaging techniques, such as magnetic resonance imaging (MRI), positron emission tomography (PET), and functional MRI (fMRI), allow researchers to non-invasively visualize and study the living brain. These techniques enable the investigation of brain abnormalities and alterations associated with various neurological and psychiatric disorders, providing crucial information for diagnosis, treatment planning, and monitoring responses to treatment. Additionally, neuroimaging provides a means to identify biomarkers, which can serve as objective measures of disease presence, progression, and treatment outcomes. By utilizing modern imaging techniques, researchers can better understand the underlying mechanisms behind brain disorders, evaluate the efficacy of interventions, and develop personalized treatment approaches. 

The considerable biological similarities between pigs and humans make pigs a highly sought-after animal model for neuroimaging studies. Pig brains are anatomically similar to human brains (Figure 4), and the large body size of the pig allows researchers to utilize clinically available equipment and techniques to study various neurological conditions. The function of the pig brain is also remarkably similar to humans. Through resting-state fMRI (rs-fMRI), Simchick et al. (2019) demonstrated that pigs have homologous resting state networks similar to humans [53]. These networks include executive control, cerebellar, sensorimotor, visual, auditory, and default mode networks. These similarities can help researchers to better understand how different disease states may affect the functional connectivity of the brain. As such, Diffusion Tensor Imaging (DTI) was used to assess fractional anisotropy in pigs with hypoxic ischemia, as published by Lee et al. (2021). Specifically, they found that DTI could be used as a diagnostic tool for identifying pigs with more swollen astrocytes in the striatum [54]. Because of the considerable similarities that the pig brain shares with the human brain, pigs serve as an excellent resource for testing the efficacy of novel therapeutic interventions.

### 2.6. Pigs as a Model Animal for Traumatic Brain Injury Research

In translational neuroscience, pigs are most commonly used for the study of traumatic brain injury (TBI). TBI is a major public health issue throughout the world, affecting millions of people annually and contributing to significant morbidity and mortality rates [55]. The Center for Disease Control and Prevention defines TBI as any disruption in brain function caused by an external force, such as a blow or jolt to the head. The severity of TBI can range from mild, such as a concussion, to severe, which often results in coma or death [56]. The pathophysiology of TBI is complex and involves both primary and secondary injury mechanisms, including mechanical damage to brain tissue, excitotoxicity, oxidative stress, inflammation, and cellular apoptosis [57,58]. Despite extensive research, effective treatments for TBI are still lacking, highlighting the need for improved understanding of the underlying mechanisms and the development of new therapeutic approaches [59]. There has been a growing concern among TBI researchers that a major reason for the poor clinical translation of treatments for TBI may be due to the animal models used [7]. As with most neuroscience research, TBI has traditionally been studied using rodent models. While the use of rats and mice has greatly increased our understanding of the cellular and molecular mechanisms behind neural injury, there are several key differences between rodent and human brains that can make it difficult to accurately predict the effects of potential treatments in humans based on rodent studies alone. As a result, there is increasing interest in using larger, more complex animal species, such as pigs, to better model the effects of TBI in humans and improve the translational potential of TBI research.

#### Studies That Use Pigs for TBI Research

The pig has been used to model various causes of TBI insults. One common method to induce TBI is the controlled cortical impact (CCI) [60]. CCI involves a craniotomy that is performed to expose the targeted region of the brain, usually the frontal or parietal cortex under general anesthesia. A pneumatic or electromagnetic impactor device is then used to deliver a controlled impact to the exposed brain tissue. The impactor is aligned and positioned to ensure accurate and consistent delivery of the impact. Researchers can adjust parameters such as impact velocity, depth, and dwell time to control the severity of the injury [61]. Following the impact, the craniotomy is typically covered with a protective material, and the scalp incision is sutured. While this is a well-characterized method for preclinical TBI in pigs [37,61,62,63,64,65], most human brain injuries are closed-head injuries [66]. Craniotomy-based models primarily mimic open-head injuries such as impalement or gunshot wounds, which may not fully capture the complexities and mechanisms of closed-head injuries, as the CCI is typically conducted with the pig’s head secured in a stereotaxic frame, not allowing for any movement of the head during impact. Nevertheless, studies using the CCI to induce TBI have led to crucial understanding on primary and secondary injury mechanisms [67,68]. 

In contrast, rotational acceleration models in pigs involve the application of rotational forces to induce brain injury [69,70,71,72]. The procedure typically involves a specialized device that allows controlled rotational movement [69]. Under general anesthesia, the head is positioned securely to ensure accurate and consistent delivery of rotational forces. The rotational acceleration can be achieved using various methods, such as a custom-built device, a pendulum, or a rotational platform. The force and duration of rotation can be adjusted to control the severity of the injury. During rotation, the pig’s head undergoes angular acceleration, leading to the deformation and shearing of brain tissue, which mimics the rotational forces experienced during closed-head TBIs in humans. The pig’s relatively large brain size is needed to make scaled-up acceleration achievable. The rotational acceleration is a strong biomedical predictor in human TBI necessary to generate the characteristic manifestations used to diagnose severity: mild TBI with diffuse injury and no imaging abnormalities; moderate TBI with 30 min–24 h of unconsciousness; and severe TBI with over 24 h of unconsciousness. Therefore, rotational acceleration in pigs is vital for preclinical TBI studies to recreate the actual mechanisms and manifestations of human injury and bridge the translational gap in neurotrauma research.

Though less common than CCI or rotational acceleration models, pigs are also used to study the effects of blast injury to the brain [73,74,75]. The blast injury model involves exposing an anesthetized animal to a shockwave generated by an explosive or compressed gas. This model is specifically designed to simulate blast-related TBI, which occurs due to the impact of explosive forces on the brain such as those experienced by military personnel in active combat. Blast TBI is often associated with complex injury mechanisms, including primary blast waves, secondary injury from flying debris, and tertiary injury from body displacement. The blast model enables researchers to investigate the unique aspects of blast-induced brain injury, including the effects of shockwaves on brain tissue, neuroinflammation, and cognitive impairments (Table 2).

Each of these TBI models has advantages and limitations, and the choice of injury model depends on the specific research questions and objectives. The different models play crucial roles in elucidating the pathophysiology of TBI, exploring therapeutic interventions and advancing our understanding of the mechanisms underlying brain injury. 

Nevertheless, there are important considerations regarding the effects of TBI when conducting research on pigs. Specifically, there is a species-specific complication that is well known in the swine industry, porcine stress syndrome (PSS). PSS is a hereditary disorder that affects pigs, particularly Landrace and Yorkshire breeds [76]. This condition can have detrimental effects on imaging and behavioral data as PSS is characterized by a hypermetabolic response and is triggered by stressors such as handling or transport. Care must be taken to reduce stress in animals susceptible to PSS.

### 2.7. Considerations for Conducting Pig Research

Pigs can serve as an incredibly beneficial intermediary model species to bridge the gap between small animal research and clinical studies. While there are many benefits to using pigs in research, there are also several considerations that need to be addressed when designing studies that will use pigs. 

Financial cost: The costs associated with using pigs for research can be much higher compared to small laboratory animals such as rodents. The initial costs to acquire pigs can vary depending on the specific breed, age, and supplier. Pigs bred specifically for research tend to be more expensive than those bred for agricultural purposes. Additionally, pig welfare is regulated under the United States Department of Agriculture, thus requiring specialized housing facilities with adequate space, ventilation, temperature control, and waste management systems. The overall size and difficulty of working with pigs can necessitate specially trained animal care staff to handle daily care such as feeding, health monitoring, and handling. Veterinary services such as routine health checkups, vaccinations, and treatment of any health issues should also be considered when budgeting for pig research. Further, conducting research with pigs often requires specialized supplies, equipment, and instrumentation. Often, clinical surgical tools and imaging equipment can be used; however, behavioral testing equipment generally must be specially engineered.

Lack of commercially available equipment: At present, there is limited availability of commercially available research supplies designed for working with pigs. Pigs are used in a wide range of research areas, and the various research applications require specialized supplies tailored to specific research objectives. The limited number of researchers who currently work with pigs and the diversity of research needs result in a lack of incentive for market suppliers to generate equipment for pig research. Additionally, pigs come in various sizes depending on breed and age. This size variability also makes it challenging to develop standardized research supplies that can accommodate the different sizes of the animals. Researchers often need to rely on adapting or modifying supplies used for other animal species, such as sheep or dogs. It is not unusual for research groups to commission customized equipment from vendors or to engage in an in-house fabrication of supplies. 

Limited genetic tools: While pigs are becoming increasingly popular as a model organism in many fields of research, the genetic tools for working with pigs are limited compared to rodent species. Pigs have a larger and more complex genome than rats and mice, which can pose a challenge for genetic manipulation. 

While these considerations and limitations exist, pigs offer unique advantages as a model organism for certain research questions, particularly in areas that require larger animal models with a closer resemblance to human physiology and behavior. Researchers must carefully weigh these factors and design experiments accordingly to maximize the benefits and minimize the limitations of using pigs for research purposes.

## 3. Discussion

Pigs have attracted much attention as a highly valuable model organism within the field of neuroscience. Pigs provide unique advantages and challenges for researchers interested in studying aspects of neural injury and disease that cannot be fully replicated in small animal models. Widely regarded for their similarities to humans in terms of neuroanatomy and neurodevelopment, pigs can help bridge the gap between small animal preclinical neuroscience and clinical applications. Pigs are particularly useful for studying cognition, behavior, and physiology in a variety of human health and disease states. Here, we have touched upon many of the commonly used behavioral tests, which have been conducted in pigs to demonstrate their intelligence and cognitive complexity. The range of relevant behaviors that pigs engage in can help researchers to identify more subtle differences in how a disease may affect an individual. Many easily replicable tests have been developed for use in pigs, and with the advancement of technology, more complex and integrative assessments are being conceptualized. Pigs are highly social animals who engage in their own forms of communication, social hierarchies, and emotional responses. The size of the pig allows for the usage of various wearable and implantable physiological monitoring devices, and neuroimaging studies can be conducted using the same equipment found in a clinical setting. Pigs are becoming one of the most sought-after model organisms for studying neurotrauma, as there are many aspects of brain injury that are unable to be replicated in a lissencephalic model species. 

## Figures and Tables

**Figure 1 biomedicines-11-02165-f001:**
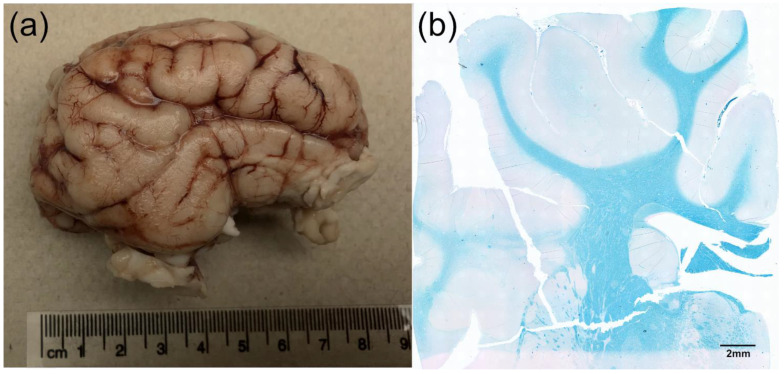
The gross anatomy of the pig brain shows similarities to human brains. (**a**) Photograph of an adult Yucatan minipig brain in sagittal view. This image shows the gross gyrencephalic anatomy of the pig cortex, depicting the presence of cerebral convolutions (gyri and sulci). (**b**) A 10× image of 5 µm thick brain slice stained for Luxol fast blue showing considerable white matter tracts in the Yucatan minipig cortex. Dark areas indicate the presence of myelin.

**Figure 2 biomedicines-11-02165-f002:**
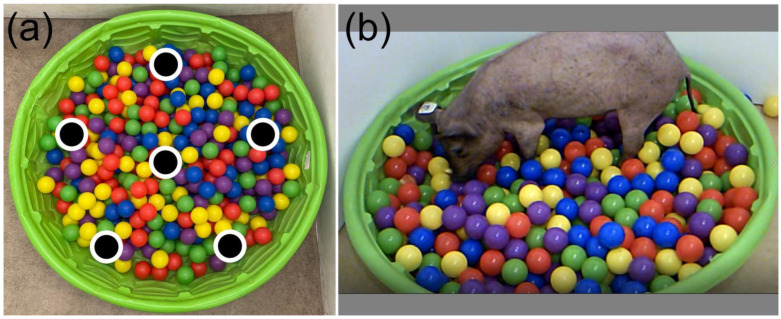
The baited ball pit—a novel spatial test to assess cognition and memory in pigs. (**a**) An overhead view of the apparatus and placement of food rewards (black circles with white outlines). (**b**) Photo showing a Yucatan minipig engaged in the ball pit task.

**Figure 3 biomedicines-11-02165-f003:**
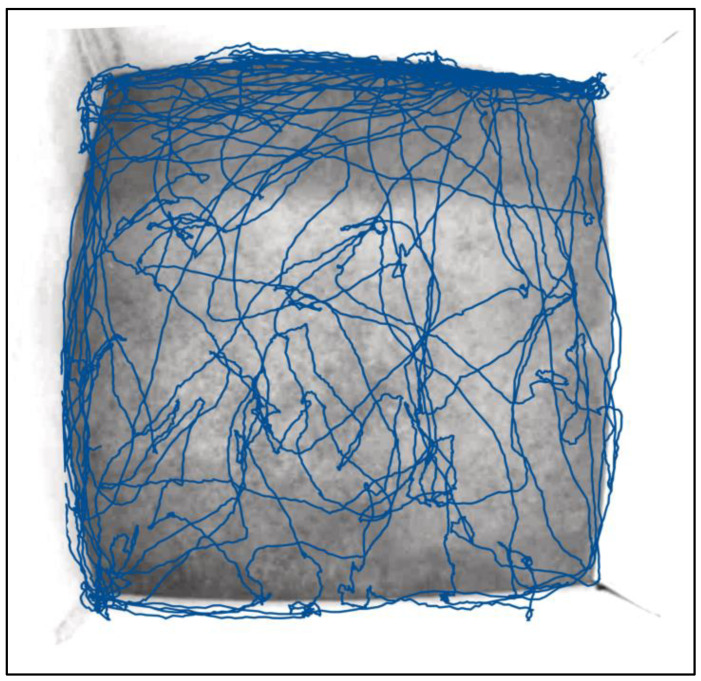
Example of pig locomotor activity in the open field. Coordinates were taken from the top of the head of a healthy Yucatan minipig over the course of 10 min via overhead video monitoring. Observations indicate that this individual spent much of their time rooting along the walls near the entrance to the chamber (top left corner). This can indicate some level of anxiety and the desire to escape the chamber.

**Figure 4 biomedicines-11-02165-f004:**
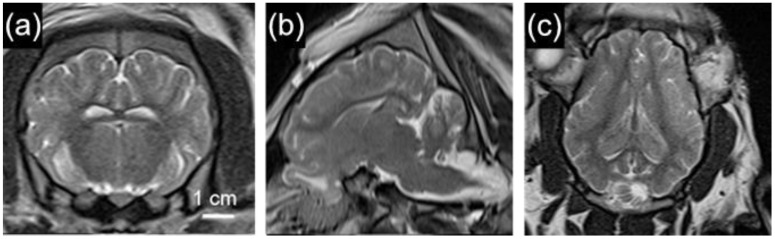
T2-weighted anatomical MR images of a minipig brain showing (**a**) coronal slice, (**b**) sagittal slice, and (**c**) axial slice. Gyri and sulci are clearly visible as well as regions of white matter.

**Table 1 biomedicines-11-02165-t001:** Summative table of highlighted behavioral tests used in pigs.

Type of Test	Authors	Experimental Condition
Maze Tasks	Kinder et al., 2019 [37]	Traumatic Brain Injury
	Singh et al., 2019 [38]	Hypoxia-Ischemia
	Chen et al., 2021 [39]	Diet
Spatial and Hole board	Clouard et al., 2022 [42]	Diet
	Netzley et al., 2021 [43]	Healthy
Object Discrimination	Schramke et al., 2016 [44]	Huntington’s Disease
	Ao et al., 2022 [45]	Healthy
Socialization	Wegner et al., 2020 [46]	Healthy
	Schramke et al., 2016 [44]	Huntington’s Disease
Vocalization	Wu et al., 2022 [47]	Healthy
	Briefer et al., 2022 [48]	Healthy
Open Field	Haigh et al., 2020 [49]	Tail biting

**Table 2 biomedicines-11-02165-t002:** Types of approaches used to model TBI in pigs.

Input Methods	Authors
Controlled Cortical Impact	Kinder et al., 2019 [37]
	Simchick et al., 2021 [62]
	Baker et al., 2019 [63]
	Manley et al., 2006 [64]
	Wang et al., 2023 [65]
Rotational Acceleration	Cullen et al., 2016 [69]
	Mayer et al., 2021 [71]
	Mayer et al., 2022 [70]
	O’Donnell et al., 2023 [72]
Blast Injury	Chen et al., 2017 [73]
	Kallakuri et al., 2017 [74]
	Cralley et al., 2022 [75]

## Data Availability

All data supporting reported results are available upon request from the corresponding author.

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
