# Peer review of "The Pig as a Translational Animal Model for Biobehavioral and Neurotrauma Research"

_biomedicines, 2023, doi:10.3390/biomedicines11082165_

Round 1

Reviewer 1 Report

This is a fantastic well written and organized timely article. I only have very minor suggestions to improve the content for the reader.

1. include several references on failure to translate preclinical to clinical TBI research

2. Briefly expand discussion on the tau accumulation, gliosis or other differences in gyrencephalic versus lissencephalic brains. 

3. Please add a brief section at the end regarding potential cost/benefit analysis of the porcine model including, but not limited to costs, specialized staff, facilities, animal ethics etc.

Author Response

Thank you for providing excellent feedback for the manuscript “The Pig as a Translational Animal Model for Biobehavioral and Neurotrauma Research”. We greatly appreciate the time and effort you have dedicated to evaluating our work. We have carefully considered each of your comments and suggestions, and tried to address all of them in the revised manuscript. We believe that this led to significant improvement to the quality and clarity of the manuscript.

Reviewer 1:

  • Comment: include several references on failure to translate preclinical to clinical TBI research

Response: Thank you for these suggestions. We have provided additional insight into the barriers of translation. Specifically, we added a new section: “Translational Considerations” that includes several new references.

  • Comment: Briefly expand discussion on the tau accumulation, gliosis or other differences in gyrencephalic versus lissencephalic brains. 

Response: We have included more details on the differences between gyrencephalic and lissencephalic brains including the differences in white matter ratio and response to mechanical stress. Specifically: “Anatomical differences between rodent and human brains are considerable. Rats and mice have small lissencephalic brains with a low white to gray matter ratio (6). In this regard, rodents may be a less than ideal model organism for the study of diseases which have a major effect on white matter regions such as Multiple Sclerosis (18). Additionally, the lack of cerebral convolutions can be problematic for studies of brain injury as stress generated by mechanical insult tends to be distributed more evenly across surface areas in smooth brains (19). In contrast, the presence of gyri and sulci in the human brain focuses mechanical stress toward the base of the sulci, deeper into the center of the brain and in close proximity to white matter regions (19).”

  • Comment: Please add a brief section at the end regarding potential cost/benefit analysis of the porcine model including, but not limited to costs, specialized staff, facilities, animal ethics etc.

Response: Thank you for this excellent suggestion. We have added a section which covers considerations researchers will need to keep in mind when planning experiments which may utilize pigs. “Pigs can serve as an incredibly beneficial intermediary model species to bridge the gap between small animal research and clinical studies. While there are many benefits to using pigs in research, there are also several considerations which need to be addressed when designing studies which will use pigs.

               Financial Cost: The costs associated with using pigs for research can be much higher compared to small laboratory animals such as rodents. Initial costs to acquire pigs can vary depending on the specific breed, age, and supplier. Pigs bred specifically for research tend to be more expensive than those bred for agricultural purposes. Additionally, pig welfare is regulated under the United States Department of Agriculture, thus requiring specialized housing facilities with adequate space, ventilation, temperature control, and waste management systems. The overall size and difficulty of working with pigs can necessitate a specially trained animal care staff to handle daily care such as feeding, health monitoring, and handling. Veterinary services such as routine health checkups, vaccinations, and treatment of any health issues should also be considered when budgeting for pig research. Further, conducting research with pigs often requires specialized supplies, equipment, and instrumentation. Often clinical surgical tools and imaging equipment can be used; however, behavioral testing equipment generally must be specially engineered.

               Lack of commercially available equipment: At present, there is limited availability of commercially available research supplies designed for working with pigs. Pigs are used in a wide range of research areas and the various research applications require specialized supplies tailored to specific research objectives. The limited number of researchers who currently work with pigs, and the diversity of research needs results in a lack of incentive for market suppliers to generate equipment for pig research. Additionally, pigs come in various sizes depending on breed and age. This size variability also makes it challenging to develop standardized research supplies that can accommodate the different sizes of the animals. Researchers often need to rely on adapting or modifying supplies used for other animal species, such as sheep or dogs. It is not unusual for research groups to commission customized equipment from vendors, or to engage in in-house fabrication of supplies.

               Limited genetic tools: While pigs are becoming increasingly popular as a model organism in many fields of research, genetic tools for working with pigs are limited compared to rodent species. Pigs have a larger and more complex genome than rats and mice which can pose a challenge for genetic manipulation.

While these considerations and limitations exist, pigs offer unique advantages as a model organism for certain research questions, particularly in areas that require larger animal models with closer resemblance to human physiology and behavior. Researchers must carefully weigh these factors and design experiments accordingly to maximize the benefits and minimize the limitations of using pigs for research purposes.”

Reviewer 2 Report

The topic of this paper is relevant, informative and of interest to the audience of this journal. This manuscript is driven by the argument that pigs share many anatomical, physiological, and genetic similarities with humans, making them valuable models for studying various aspects of human health and disease. As such the similarities in the size and structure of the pig brain (and spinal cord) to those of humans make them suitable models for studying the mechanisms of injury, evaluating treatment strategies, and assessing functional outcomes. Pigs allow researchers to mimic and study the effects of TBI in a controlled laboratory environment, facilitating the development of interventions and therapies. Pigs also offer advantages in terms of surgical procedures and monitoring capabilities. Their larger size compared to traditional small animal models like rodents allows for more precise surgical interventions and easier monitoring of physiological parameters. Moreover, it should be stated that large animal models (eg: pig) may present both a financial and logistical restrictions that may impact the statistical factor due to their small sample size that is most often employed in the use of large animal models for preclinical research purposes.

I have only five issues, which the authors might want to highlight and expand further:

1.      The first issue is that the authors did not elaborate on the fundamental importance in the use of large animal models (such as the pig) to evaluate for the proper efficacy and long term or negative interactions of drugs including dosage. We all know that neuroprotective drugs that work well on rodents have shown disappointing results in humans. This is the classical case of NMDA antagonists.  Reason being that rodents tolerate larger drug doses that would otherwise have a serious cardiovascular effects and others on humans.

2.      Secondly, one must also mention the limited commercial availability of molecular tools and transgenics in large animal models unlike rodents and build around the argument that small experimental animals would ideally be used as exploratory tools to test for hypothesis/mechanisms rather than preclinical, end-stage testing.

3.      Thirdly, porcine stress syndrome that is marked by increased body temperature caused by stress, and specific anesthetic agents have been often shown conflicting physiological, imaging and behavioural data during neuromonitoring in traumatic conditions such as in TBI and similar. It would be of great value to the readership of this manuscript if the authors addressed and expanded this major issue with reference to published physiological/methodological data in pigs in models of TBI with drugability centred studies and presented precautions and recommendations.

4.      Fourthly, a table summarizing the advantages and disadvantages of the use of the pig as a model for behavioral/neurotrauma research vis a vis the commonly used rodent models would be extremely useful for ‘at a glance read’ to gather more focus and attention.

5.      Fifth. The manuscript would be more complete if it is accompanied by some important published physiological data derived from neurotrauma in pigs to amplify on the extensive monitoring data that could be derived in the use of such a model in TBI etc and how this data is similar to that found in humans (edema formation, neurotransmitter release, EEG, ROS etc).

Overall, the paper is easy to follow, clear and free from grammatical or spelling errors. The review is informative and well planned and with good quality of English language useage and grammar. The topic can be considered to be an emerging area and provides a critical and constructive analysis of the literature whilst providing valuable insights in large animal and which are mostly used in confirmative studies. I am sure that following my recommendations, the paper will be further strengthened.

Author Response

Thank you for providing excellent feedback for the manuscript “The Pig as a Translational Animal Model for Biobehavioral and Neurotrauma Research”. We greatly appreciate the time and effort you have dedicated to evaluating our work. We have carefully considered each of your comments and suggestions, and tried to address all of them in the revised manuscript. We believe that this led to significant improvement to the quality and clarity of the manuscript.

Reviewer 2:

2.1. Comment: elaborate on the fundamental importance in the use of large animal models (such as the pig) to evaluate for the proper efficacy and long term or negative interactions of drugs including dosage.

Response: Thank you for pointing out this important aspect. We have added details from a recent article showing that expression of genes for neurotransmitter receptors and ion channels in rodents are different from humans. Specifically: “A recent report by Hodge et al (2019) has demonstrated considerable differences in the expression of genes among similar cell types within mouse and human brains (22). In this study, researchers used single-cell transcriptomics to characterize various cell types in the cortex of mice and human middle temporal gyrus. They then compared the expression of genes in homologous cell types and found that among similar cell types, there are major divergencies in the expression of neurotransmitter receptors, ion channels, extracellular matrix elements, and cell-adhesion molecules between species (22).”

2.2. Comment: mention the limited commercial availability of molecular tools and transgenics in large animal models

Response: Thank you for this valuable suggestion. We have summarized barriers to large animal use including limited availability of equipment and genetic tools. Specifically, we added a section” Considerations for Conducting Pig Research”: Financial Cost: The costs associated with using pigs for research can be much higher compared to small laboratory animals such as rodents. Initial costs to acquire pigs can vary depending on the specific breed, age, and supplier. Pigs bred specifically for research tend to be more expensive than those bred for agricultural purposes. Additionally, pig welfare is regulated under the United States Department of Agriculture, thus requiring specialized housing facilities with adequate space, ventilation, temperature control, and waste management systems. The overall size and difficulty of working with pigs can necessitate a specially trained animal care staff to handle daily care such as feeding, health monitoring, and handling. Veterinary services such as routine health checkups, vaccinations, and treatment of any health issues should also be considered when budgeting for pig research. Further, conducting research with pigs often requires specialized supplies, equipment, and instrumentation. Often clinical surgical tools and imaging equipment can be used; however, behavioral testing equipment generally must be specially engineered.

               Lack of commercially available equipment: At present, there is limited availability of commercially available research supplies designed for working with pigs. Pigs are used in a wide range of research areas and the various research applications require specialized supplies tailored to specific research objectives. The limited number of researchers who currently work with pigs, and the diversity of research needs results in a lack of incentive for market suppliers to generate equipment for pig research. Additionally, pigs come in various sizes depending on breed and age. This size variability also makes it challenging to develop standardized research supplies that can accommodate the different sizes of the animals. Researchers often need to rely on adapting or modifying supplies used for other animal species, such as sheep or dogs. It is not unusual for research groups to commission customized equipment from vendors, or to engage in in-house fabrication of supplies.

               Limited genetic tools: While pigs are becoming increasingly popular as a model organism in many fields of research, genetic tools for working with pigs are limited compared to rodent species. Pigs have a larger and more complex genome than rats and mice which can pose a challenge for genetic manipulation.

2.3.      Thirdly, porcine stress syndrome that is marked by increased body temperature caused by stress, and specific anesthetic agents have been often shown conflicting physiological, imaging and behavioral data during neuromonitoring in traumatic conditions such as in TBI and similar. It would be of great value to the readership of this manuscript if the authors addressed and expanded this major issue with reference to published physiological/methodological data in pigs in models of TBI with drugability centered studies and presented precautions and recommendations.

Response: Thank you for this suggestion. Porcine stress syndrome is an interesting condition and not one that we have found much literature on. We have included PSS as an important consideration when considering pig research, however, peer reviewed publications on the subject are limited and not many have been published in recent years.

2.4.      Fourthly, a table summarizing the advantages and disadvantages of the use of the pig as a model for behavioral/neurotrauma research vis a vis the commonly used rodent models would be extremely useful for ‘at a glance read’ to gather more focus and attention.

Response: Thank you for this great suggestion. We now added two sections, specifically,” Preclinical Research” and Translational Considerations” that dive in depth on the advantages and disadvantages of each of these animal models. These sections include many new and important references.

2.5.      Fifth. The manuscript would be more complete if it is accompanied by some important published physiological data derived from neurotrauma in pigs to amplify on the extensive monitoring data that could be derived in the use of such a model in TBI etc and how this data is similar to that found in humans (edema formation, neurotransmitter release, EEG, ROS etc).

Response: Thank you for this important suggestion. We have included references discussing the formation and monitoring of post traumatic epilepsy via EEG in pigs in the section entitled “real-time physiological monitoring”. We have also discussed changes in fractional anisotropy in pigs with ischemic injuries using DTI.

Reviewer 3 Report

The authors provide a general narrative description of how pigs are better suited for cognitive research and for modelling craniocerebral injuries than rodent models. Facts well-known and evident to researchers are listed based on an incomplete literature review. What is the purpose of the narrative summation, to provide academic assistance to TBI experimental researchers? In the latter case, it would be reasonable to demonstrate with a specific example how the pig experiment offers more than the rodent model. This paper does not provide the reader with additional information that a simple literature search could not immediately reach. The paper should have some focus to arouse the reader's interest!

Author Response

Thank you for providing excellent feedback for the manuscript “The Pig as a Translational Animal Model for Biobehavioral and Neurotrauma Research”. We greatly appreciate the time and effort you have dedicated to evaluating our work. We have carefully considered each of your comments and suggestions, and tried to address all of them in the revised manuscript. We believe that this led to significant improvement to the quality and clarity of the manuscript.

Reviewer 3:

Comment: The authors provide a general narrative description of how pigs are better suited for cognitive research and for modelling craniocerebral injuries than rodent models. Facts well-known and evident to researchers are listed based on an incomplete literature review. What is the purpose of the narrative summation, to provide academic assistance to TBI experimental researchers? In the latter case, it would be reasonable to demonstrate with a specific example how the pig experiment offers more than the rodent model. This paper does not provide the reader with additional information that a simple literature search could not immediately reach. The paper should have some focus to arouse the reader's interest!

Response: Thank you for your feedback. We have carefully considered your comment and revised the paper in a way that will provide a unique view of pig research. To do so, we have included three new sections, over 20 references, and also figures that describe results of our lab research, that had never been published previously. The new sections are: “Preclinical Research”, Translational Consideration”, and “Considerations for Conducting Pig Research”.  We believe the revised manuscript, that demonstrate how pig research may be relevant to biomedical research in ways that rodents research can not be due to anatomical and genetic considerations, provides a valuable collection of references, new interpretation, and a collection of tools that have not been shared before in such a comprehensive way.

Round 2

Reviewer 3 Report

The revised manuscript demonstrates how pig research may be relevant to biomedical research in ways that rodents research can not be due to anatomical and genetic considerations! It provides a valuable collection of references, new interpretation, and a collection of tools that have not been shared before in such a comprehensive way!

Author Response

Thank you.